# KARA: Enhancing High-Dimensional Data Processing with Learnable Activations

## Abstract

In the rapidly advancing field of machine learning, efficiently processing and interpreting high-dimensional data remains a significant challenge. This paper presents the Kolmogorov-Arnold Representation Autoencoder (KARA), a novel autoencoder architecture designed to leverage the Kolmogorov-Arnold representation theorem. By incorporating this mathematical foundation, KARA enhances the representational power and efficiency of neural networks, enabling superior performance in data compression tasks. Experimental results demonstrate that KARA achieves superior performance, positioning it as a promising approach for high-dimensional data processing.

## 1 Introduction

In the era of big data, machine learning algorithms are increasingly tasked with processing and interpreting high-dimensional datasets across various domains (LeCun et al., 2015; Dong et al., 2021). High-dimensional data, while rich in information, pose significant challenges related to computational complexity, storage requirements, and the risk of overfitting. Efficiently capturing the underlying structure of such data is crucial for enhancing the performance of machine learning models and enabling their deployment in resource-constrained environments (Chen & Ran, 2019).

Recent advancements, such as the success of Kolmogorov-Arnold Networks (KAN) (Liu et al., 2024), have opened promising avenues for augmenting the capacity of multilayer perceptrons (MLPs), particularly within the realms of mathematics and physics. Building upon this foundation, we introduce Kolmogorov-Arnold Representation Autoencoders (KARA), a novel framework specifically designed to address the intricacies of high-dimensional data. KARA incorporates learnable activation functions within the autoencoder architecture, distinguishing itself from traditional models that rely on static activation functions. This dynamic adaptation of activation functions allows KARA to tailor its representations based on the underlying data, thereby offering a more flexible and potent modeling capability.

Through experimentation on image autoencoding tasks, we demonstrate that KARA achieves superior performance, highlighting its potential as a robust solution for complex data modeling tasks. Our findings suggest that pixel data can be effectively encoded through combinations of smooth functions, paving the way for more efficient and accurate representations in high-dimensional spaces.

## 2 Related Work

**Cox-de Boor formula.** The Cox-de Boor formula provides an efficient, recursive method for computing B-spline basis functions. This formula is fundamental in the construction of smooth and flexible spline curves (Bohra et al., 2020). The Cox-de Boor recursion is defined by the following equations:

$$B_{i,0}(x) = \begin{cases} 1, & \text{if } x_i \leq x < x_{i+1}, \\ 0, & \text{otherwise,} \end{cases} \tag{1}$$

$$B_{i,p}(x) = \frac{x - x_i}{x_{i+p} - x_i} B_{i,p-1}(x) + \frac{x_{i+p+1} - x}{x_{i+p+1} - x_{i+1}} B_{i+1,p-1}(x), \tag{2}$$

where $B_{i,p}(x)$ represents the B-spline basis function of degree $p$, associated with the knot sequence $\{x_i\}$. This recursive formulation ensures a specified degree of smoothness and continuity in the

resulting spline curve (Ahlberg et al., 2016). In our proposed KARA, the Cox-de Boor formula is employed to generate smooth and continuous representations of high-dimensional data.

**Kolmogorov-Arnold representation theorem.** The Kolmogorov-Arnold representation theorem posits that any continuous function mapping from $[0,1]^n$ to $\mathbb{R}$, denoted by $f$, can be expressed as a finite composition of continuous univariate functions, combined through addition operations (Schmidt-Hieber, 2021). This theorem is foundational in understanding high-dimensional function approximation and is formally stated as follows:

$$f(\mathbf{x}) = f(x_1, \dots, x_n) = \sum_{q=1}^{2n+1} \Phi_q \left( \sum_{p=1}^{n} \psi_{q,p}(x_p) \right), \tag{3}$$

where each $\psi_{q,p} : [0,1] \to \mathbb{R}$ and $\Phi_q : \mathbb{R} \to \mathbb{R}$ are continuous univariate functions. KAN leverage this theorem for efficient function fitting (Liu et al., 2024). Our proposed KARA further exploits this theoretical framework to enhance performance in processing high-dimensional data, particularly in tasks such as dimensionality reduction and data reconstruction.

## 3 METHOD

### 3.1 ARCHITECTURE

The architecture of KARA is designed to effectively handle high-dimensional data through a structured transformation process. The encoder and decoder components are central to this process, functioning to reduce dimensionality and subsequently reconstruct the original data.

The encoder part of KARA transforms high-dimensional input data $\mathbf{x}$ into a compressed, lower-dimensional latent space $\mathbf{z}$. This transformation is achieved through a series of function compositions involving multiple layers:

$$\mathbf{z} = (\Phi_{L-1} \circ \Phi_{L-2} \circ \cdots \circ \Phi_0)(\mathbf{x}). \tag{4}$$

Each layer $\Phi_i$ represents a specific transformation stage, contributing to the progressive encoding of the input data into more abstract representations. The final output $\mathbf{z}$ serves as the encoded latent representation, capturing the essential features of the input necessary for reconstruction.

Conversely, the decoder part of KARA is responsible for reconstructing the original data from its latent representation $\mathbf{z}$. Similar to the encoder, the decoder consists of a series of layers, each of which progressively reconstructs the higher-dimensional data from the encoded state:

$$\hat{\mathbf{x}} = (\Psi_{L-1} \circ \Psi_{L-2} \circ \cdots \circ \Psi_0)(\mathbf{z}). \tag{5}$$

Each layer $\Psi_i$ performs a specific function that gradually transforms the latent variables back to a state $\hat{\mathbf{x}}$ that closely resembles the original input data $\mathbf{x}$.

### 3.2 LEARNABLE ACTIVATION LAYER

The Learnable Activation (LA) layer employs a set of learnable activation functions, collectively denoted as $\Phi$. This configuration allows the LA layer to dynamically adjust during the training process. The activation for an input vector $\mathbf{x}$ is mathematically defined as:

$$\Phi(\mathbf{x}) = \sum_{i=1}^{N} \phi_i(x_i), \tag{6}$$

where each function $\phi_i(x_i)$ is specifically designed to optimize particular attributes of the input data, thereby enhancing the overall efficacy of the layer.

Each individual activation function $\phi(x)$ within this series incorporates a spline-based approach coupled with a foundational activation function, offering a robust mechanism for handling non-linear data transformations:

$$\phi(x) = \sum_{i=1}^{l+p} B_{i,p}(x) \cdot \mathbf{w}_i + b(x) \cdot \mathbf{w}_0, \tag{7}$$

where $B_{i,p}(x)$ are B-spline basis functions of degree $p$, $l$ represents the grid size, $\mathbf{w}_i$ are the corresponding trainable coefficients, and $b(x)$ represents the base activation function scaled by the trainable vector $\mathbf{w}_0$.

### 3.3 SHIFT-INVARIANT LEARNABLE ACTIVATION LAYER

Inspired by the architecture of Convolutional Neural Networks (CNNs) (LeCun et al., 1998), the Shift-Invariant Learnable Activation (SILA) layer is designed to introduce shift-invariant properties within neural networks, enabling the model to capture patterns irrespective of their spatial position. Unlike the Learnable Activation (LA) layer, which requires a separate set of parameters for each input dimension, the SILA layer leverages parameter sharing across different input locations, significantly reducing the number of trainable parameters.

The kernel of the SILA layer consists of learnable activation functions organized in a matrix format:

$$\Phi = \begin{bmatrix} \phi_{1,1} & \phi_{1,2} & \cdots & \phi_{1,m} \\ \phi_{2,1} & \phi_{2,2} & \cdots & \phi_{2,m} \\ \vdots & \vdots & \ddots & \vdots \\ \phi_{n,1} & \phi_{n,2} & \cdots & \phi_{n,m} \end{bmatrix}, \tag{8}$$

where each element $\phi_{i,j}$ is a distinct learnable activation function, and $n$ and $m$ define the dimensions of the kernel.

The application of this matrix to an input vector $\mathbf{x}$ yields a composite output, calculated as follows:

$$\Phi(\mathbf{x}) = \begin{bmatrix} \sum_{i=1}^{n}\sum_{j=1}^{m}\phi_{i,j}(x_{i,j}) & \sum_{i=1}^{n}\sum_{j=1}^{m}\phi_{i,j}(x_{i,j+s}) & \cdots & \sum_{i=1}^{n}\sum_{j=1}^{m}\phi_{i,j}(x_{i,j+qs}) \\ \sum_{i=1}^{n}\sum_{j=1}^{m}\phi_{i,j}(x_{i+s,j}) & \sum_{i=1}^{n}\sum_{j=1}^{m}\phi_{i,j}(x_{i+s,j+s}) & \cdots & \sum_{i=1}^{n}\sum_{j=1}^{m}\phi_{i,j}(x_{i+s,j+qs}) \\ \vdots & \vdots & \ddots & \vdots \\ \sum_{i=1}^{n}\sum_{j=1}^{m}\phi_{i,j}(x_{i+ps,j}) & \sum_{i=1}^{n}\sum_{j=1}^{m}\phi_{i,j}(x_{i+ps,j+s}) & \cdots & \sum_{i=1}^{n}\sum_{j=1}^{m}\phi_{i,j}(x_{i+ps,j+qs}) \end{bmatrix} \tag{9}$$

where $s$ is the stride and $p$ and $q$ represent the spatial dimensions of the output matrix, respectively.

### 3.4 SPARSIFICATION

Sparsification in neural networks is a key technique used to reduce model complexity and improve interpretability by encouraging sparse connections between neurons (Louizos et al., 2017). In our KARA framework, this sparsification is quantified using the L1 norm and the entropy of activation functions within the layers.

The L1 norm of an activation function $\phi$ is defined to reflect its sparsity and is given by:

$$|\phi|_1 = \frac{1}{l+p}\sum_{i=1}^{l+p}|\mathbf{w}_i|, \tag{10}$$

where $p$ denotes the degree of the spline, and $\mathbf{w}_i$ represents the weights associated with $\phi$.

For a LA layer, the L1 norm is calculated as the sum of the L1 norms of all its activation functions:

$$|\Phi|_1 = \sum_{i=1}^{n}|\phi_i|_1. \tag{11}$$

The entropy of a layer $\Phi$, which measures the uniformity of the distribution of activation sparsities, is defined as:

$$S(\Phi) = -\sum_{i=1}^{n}\frac{|\phi_i|_1}{|\Phi|_1}\log\left(\frac{|\phi_i|_1}{|\Phi|_1}\right). \tag{12}$$

Similarly, for a SILA layer, the L1 norm is computed as:

$$|\Phi|_1 = \sum_{i=1}^{n} \sum_{j=1}^{m} |\phi_{i,j}|_1. \tag{13}$$

The entropy is defined as:

$$S(\Phi) = -\sum_{i=1}^{n} \sum_{j=1}^{m} \frac{|\phi_{i,j}|_1}{|\Phi|_1} \log \left( \frac{|\phi_{i,j}|_1}{|\Phi|_1} \right). \tag{14}$$

The total training objective $\ell$ incorporates the prediction loss along with L1 and entropy regularization for all layers:

$$\ell = \|\mathbf{x} - \hat{\mathbf{x}}\|^2 + \lambda \left( \mu_1 \sum_{l=0}^{L-1} |\Phi_l|_1 + \mu_2 \sum_{l=0}^{L-1} S(\Phi_l) + \mu_3 \sum_{l=0}^{L-1} |\Psi_l|_1 + \mu_4 \sum_{l=0}^{L-1} S(\Psi_l) \right), \tag{15}$$

where $\mu_1$, $\mu_2$, $\mu_3$, and $\mu_4$ are coefficients typically set to 1, and $\lambda$ is a scaling factor that adjusts the overall impact of regularization. This comprehensive objective function is designed to strike a balance between maintaining fidelity to the training data and enforcing sparsity and distributional uniformity across the network's layers. This balance aids in faster convergence and improves generalization by reducing the risk of overfitting (Hoefler et al., 2021).

## 4 EXPERIMENTS

### 4.1 EXPERIMENT SETUP

**Datasets.** For our experiments, we utilize two widely recognized benchmark datasets: MNIST and Fashion-MNIST. The MNIST dataset comprises 60,000 training images and 10,000 test images, each representing handwritten digits from 0 to 9 in a standardized $28 \times 28$ grayscale pixel format (Deng, 2012). Fashion-MNIST, designed as a more challenging alternative to MNIST, shares the same data structure but contains grayscale images of 10 different types of fashion items, such as shirts, shoes, and bags, providing a more complex and diverse set of visual patterns for model evaluation (Xiao et al., 2017).

**Pre-training.** Training of our model begins with a self-supervised learning phase lasting 50 epochs. We utilize the AdamW optimizer, known for its improved weight decay handling and efficiency in large-scale training scenarios (Loshchilov & Hutter, 2017). The batch size is set to 1024, ensuring a balance between computational efficiency and gradient stability. We initialize the learning rate at 0.001 and apply a cosine annealing schedule (Loshchilov & Hutter, 2016), which progressively reduces the learning rate in a smooth, cyclic manner.

**Evaluation methodology.** The evaluation of the model's reconstruction quality is conducted using the Mean Squared Error (MSE), a widely used metric that quantifies the average squared difference between the original input and the reconstructed output. Lower MSE values indicate higher accuracy in reproducing the input data, effectively measuring the model's ability to capture intricate data patterns. To further assess the quality of the learned representations, we employ the linear probing technique, which involves training a linear classifier on the frozen features extracted from the pre-trained network. This approach evaluates the discriminative power of the learned representations by predicting the classes of images in the test set, offering insight into the effectiveness of the model's feature learning in downstream tasks (Alain & Bengio, 2016; Kornblith et al., 2019).

### 4.2 ABLATION STUDIES

To comprehensively evaluate the effectiveness of the proposed Kolmogorov-Arnold Representation Autoencoder (KARA), we conducted a series of ablation studies on MNIST dataset. These studies aim to isolate and understand the impact of key components within our model. Each ablation experiment involves systematically modifying or removing a component and observing the resultant changes in performance metrics.

Table 1: **Ablation studies.** We conducted a series of ablation experiments to evaluate the impact of key components in KARA. Default settings are highlighted in  gray .

(a) **Sparsification magnitude.**

| $\lambda$ | MSE | Acc (%) |
|---|---|---|
| $10^0$ | 0.0135 | 88.10 |
| $10^{-1}$ | 0.0113 | 89.22 |
| $10^{-2}$ | **0.0095** | **90.01** |
| $10^{-3}$ | 0.0104 | 89.56 |

(b) **Encoder design.**

| Type | MSE | Acc (%) |
|---|---|---|
| Linear | 0.0134 | 84.83 |
| Conv | 0.0134 | 84.56 |
| LA | 0.0132 | 89.92 |
| SILA | **0.0095** | **90.01** |

(c) **Decoder design.**

| Type | MSE | Acc (%) |
|---|---|---|
| Linear | 0.0104 | 89.90 |
| LA | **0.0095** | **90.01** |

Table 2: **Model robustness.** KARA shows improved performance as the latent dimensions increase, evaluated on MNIST and Fashion-MNIST datasets.

| Encoder | Decoder | Dim | MNIST | | Fashion-MNIST | |
|---|---|---|---|---|---|---|
| | | | MSE | Acc (%) | MSE | Acc (%) |
| Linear | Linear | 8 | 0.0170 | 79.38 | 0.0145 | 72.08 |
| | | 16 | 0.0134 | 82.80 | 0.0127 | 74.12 |
| | | 32 | 0.0131 | 84.65 | 0.0120 | 77.03 |
| Conv | Linear | 8 | 0.0165 | 79.13 | 0.0150 | 66.07 |
| | | 16 | 0.0135 | 81.44 | 0.0132 | 73.72 |
| | | 32 | 0.0131 | 86.08 | 0.0130 | 75.70 |
| LA | LA | 8 | 0.0135 | 88.10 | 0.0126 | 77.02 |
| | | 16 | 0.0132 | 89.92 | 0.0104 | 78.34 |
| | | 32 | 0.0132 | 90.40 | 0.0095 | 79.05 |
| SILA | LA | 8 | **0.0101** | **88.50** | **0.0112** | **78.17** |
| | | 16 | **0.0095** | **90.01** | **0.0095** | **79.92** |
| | | 32 | **0.0089** | **91.24** | **0.0077** | **81.01** |

**Sparsification magnitude.** To assess the influence of sparsification, we evaluate models with varying sparsification magnitudes. The results are presented in Table 1a. Our findings reveal that sparsification has a notable influence on model performance, affecting both the reconstruction error and the accuracy of linear probing.

**Encoder design.** To understand the role of encoder architecture, we investigate the impact of different encoder architectures on KARA's performance. The encoders compared include linear layers, convolutional layers, LA layers, and SILA layers. The performance metrics for each encoder type are detailed in Table 1b. The results demonstrate that the SILA encoder outperforms the other architectures.

**Decoder design.** To understand the role of decoder architecture, we compare the performance of decoders utilizing linear layers versus LA layers. The comparative results are shown in Table 1c. The LA decoder consistently achieves better performance metrics, indicating that learnable activation functions in the decoder contribute to more accurate data reconstruction.

## 4.3 MODEL ROBUSTNESS

We evaluate the robustness of our KARA model by varying the dimensionality of the latent space and analyzing its impact on performance across both MNIST and Fashion-MNIST datasets. As shown in Table 2, KARA exhibits improved performance as the latent dimension increases.

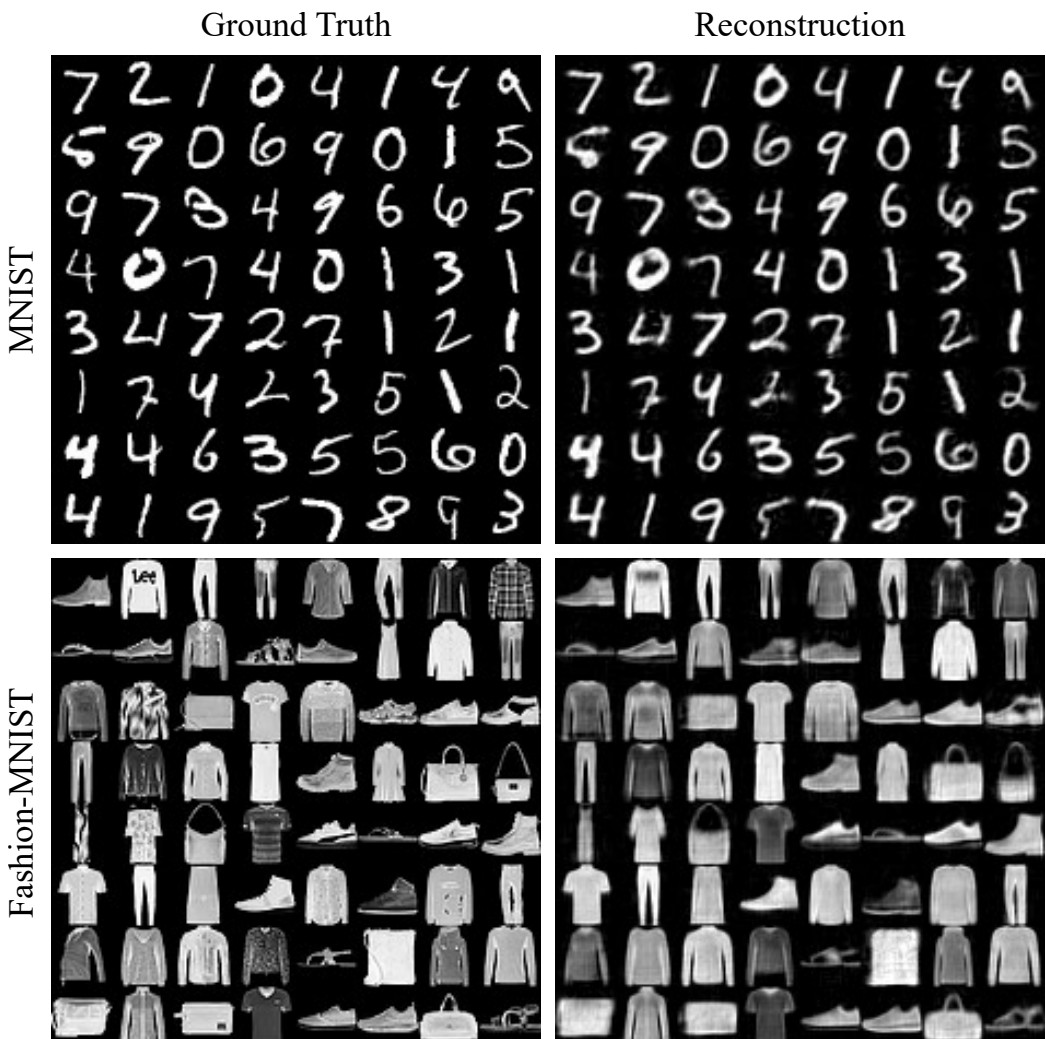

Figure 1: **Visual comparison of original and reconstructed images.** The left column displays the original images, while the right column shows the corresponding reconstructed images produced by our KARA. The top two rows illustrate samples from the MNIST dataset, and the bottom two rows present samples from the Fashion-MNIST dataset.

### 4.4 QUALITATIVE ANALYSIS

**Reconstructed images.** A visual comparison between the original images and the reconstructed images produced by our KARA model is shown in Figure 1. The KARA model demonstrates a strong ability to preserve the shapes and contours of objects.

**Latent space interpolation.** We perform uniform interpolation in the latent space, generating a smooth transition between different data points. The resulting images, decoded from the interpolated latent vectors, are displayed in Figure 2. This demonstrates the model's ability to capture meaningful variations in the latent space, reflecting continuous transformations in the underlying data distribution.

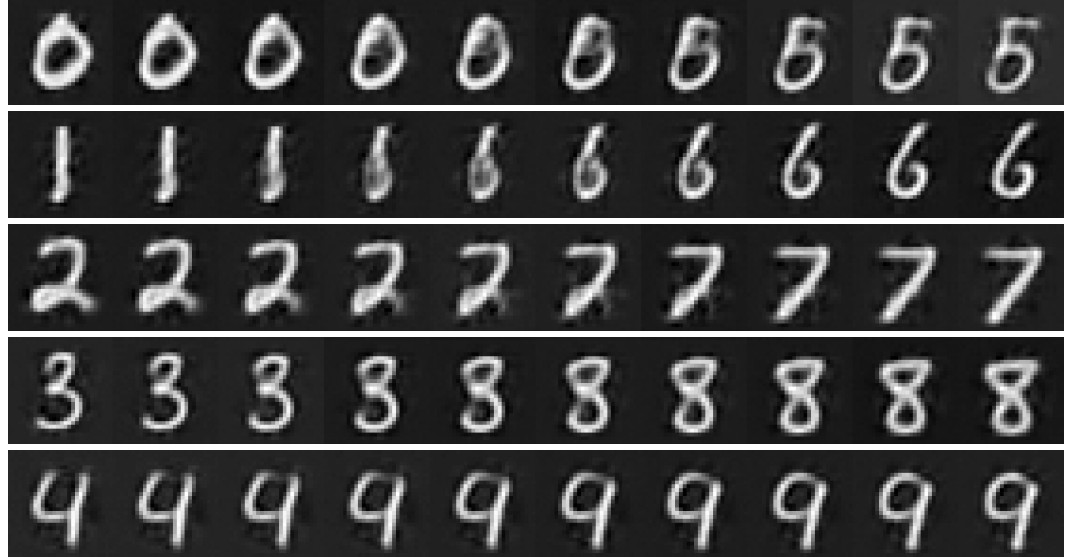

Figure 2: **Latent space interpolation.** The leftmost and rightmost columns show the original digits. The intermediate columns display the smooth transition between these digits, demonstrating how one digit gradually morphs into another through interpolation within the latent space.

## 5 DISCUSSION

The performance of our KARA model in environments with constrained latent dimensions underscores its strong potential for applications that require preserving information integrity while achieving substantial dimensionality reduction. The experimental results demonstrate that incorporating learnable activation functions is effective for tasks involving complex data representations. This suggests that learnable activations could play a critical role in improving the efficiency and flexibility of autoencoder models, aligning with ongoing advancements in autoencoder research (Li et al., 2023).

Looking ahead, future research should focus on extending the application of KARA to more complex, multi-layer network architectures. This would enable a deeper evaluation of the model's performance improvements, as well as any potential trade-offs, such as increased computational demands or overfitting risks (Goodfellow, 2016). Additionally, it will be essential to assess the computational efficiency of KARA in large-scale deployments, particularly in scenarios with massive datasets or real-time processing requirements. Evaluating its adaptability across a broader range of tasks and datasets will be crucial for determining its scalability and robustness in diverse domains.

Moreover, the application of KARA in critical fields such as healthcare, where interpretability, precision, and data efficiency are paramount, presents an exciting avenue for future exploration. In such contexts, the ability of KARA to effectively reduce data dimensionality while preserving essential features could yield substantial practical benefits (Vessies et al., 2023). Investigating its use in medical imaging, diagnostics, and other areas where both accuracy and explainability are critical could provide significant insights into its real-world potential (Esteva et al., 2021).

## 6 CONCLUSION

In conclusion, we introduced the Kolmogorov-Arnold Representation Autoencoder (KARA), a novel framework designed to address the challenges of high-dimensional data representation by integrating learnable activation functions within an autoencoder architecture. Through experiments on benchmark datasets, KARA demonstrated its ability to effectively capture complex data structures while maintaining a reduced number of parameters. The incorporation of dynamic, learnable activation functions proved to be particularly effective, enabling more flexible and efficient data encoding.

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
