# OpenReview forum: "KARA: Enhancing High-Dimensional Data Processing with Learnable Activations"
_ICLR.cc/2025/Conference — ICLR 2025 Conference Withdrawn Submission_

### Official Review · Reviewer_4P2F · 2024-10-29

**Soundness:** 2
**Presentation:** 3
**Contribution:** 1
**Rating:** 3
**Confidence:** 4

**Summary:**

The authors combine two KANs into an autoencoder and evaluate the reconstruction quality of the decoder output. They compare the performance of MLPs, CNNs, KANs and convolutional KANs for the task of encoding MNIST and Fashion MNIST.

**Strengths:**

All technical sections are easy to follow, the mathematical notation is clear. I appreciate the rather “verbose” notation in section 3.3 - the strategy is clear at first sight.

**Weaknesses:**

1. Section 3.1, 3.2 and 3.4 describe advancements presented in the original KAN paper, Section 3.3 describes convolutional KANs (https://arxiv.org/abs/2406.13155) (without reference to the corresponding paper). Combining two KANs into an Autoencoder to analyse the latent space is a viable application of KANs and yields interesting results. However, to justify a publication in a conference like ICLR, I would expect either a contribution to the model, or a more detailed analysis of the outcome.
2. The related work section does not reference any prior research on KANs, such as U-KAN (https://arxiv.org/abs/2406.02918), KAT (https://arxiv.org/abs/2409.10594), or KAN Autoencoders (https://arxiv.org/abs/2410.02077) (see also https://github.com/mintisan/awesome-kan).
3. Unfortunately, no code is provided.

**Questions:**

1. It would have been interesting to include the number of parameters into the evaluation. The SILA layer should contain notably fewer parameters than a regular LA layer. Is it also faster to train? What are the sizes of the kernels that you use for the layer, and how many of them do you combine? How many layers are used in the different architectures?
2. MNIST and FashionMNIST are comparatively simple datasets. How does KARA perform on more complex datasets (like SVHN or Cifar10)?

---

### Official Review · Reviewer_LmkY · 2024-10-30

**Soundness:** 2
**Presentation:** 3
**Contribution:** 2
**Rating:** 1
**Confidence:** 4

**Summary:**

This work proposed an autoencoder network with learnable activation functions built upon the Kolmogorov-Arnold representation theorem.
The learnable activation function is based on the Cox-de-Boor formula for computing the B-spline basis function.
This network is then tested against dimensionality reduction tasks.

**Strengths:**

The strength of the paper is as follows:

1. The paper is generally written and easy to understand.
2. The proposed method for generating learnable activation functions makes sense.
3. The experiments are valid, although limited to image dimensionality reduction tasks.

**Weaknesses:**

The primary weakness of this paper is its narrow scope. The arguments for the efficacy of the proposed method are only centered on its ability to produce better reproduction accuracies compared to other models, while potentially interesting insights on the characteristics of the produced activation functions are entirely missing.

Details of suggestions are as in the Question session of this review

**Questions:**

1. Table 2: Why is the SILA-SILA model missing from the accuracy test in Table 2? Please explain why the decoder part does not require SILA.

2. The authors demonstrated that their model produces better accuracy than other autoencoder models, but are the results significant, especially considering the cost of training the activation functions? This can be analyzed, for example, by adding a classification layer and observing how the latent space contributes to the network's classification accuracy.

3. Are the good results due to the trainability of the activation functions or just due to the mixing of different types of activation functions that can be achieved without the learning process, for example, randomly mixing fixed activation functions?
Please run additional experiments to compare the proposed network against some networks with mixed but fixed activation functions.

4. The depth of this work can be significantly improved if the authors also analyze the resulting activation functions. For example, what functions emerge from the learning process? Does the process produce significantly different activation functions for each feature? If not, how are the different activation functions distributed, etc.?
Please visualize some of the activation functions. If possible, consider measuring the diversity and distribution of the learned activation functions.

5. What are the essential aspects that shape the activation function? Is it the problem? Or the structure of the network?
Please consider using the measured diversity of the activation functions across different problem types or network architectures to identify which aspects significantly impact their shapes.

---

> ### Comment · Reviewer_LmkY · 2024-12-02
> **final comment**
>
> I am sorry that I have to decrease the rating because the authors have not responded to the review.

---

### Official Review · Reviewer_jzoo · 2024-11-02

**Soundness:** 3
**Presentation:** 3
**Contribution:** 1
**Rating:** 1
**Confidence:** 4

**Summary:**

This paper introduces a Kolmogorov-Arnold Representation Autoencoder (KARA) to process high-dimensional data. The encoder of KARA encodes the input data progressively to lower dimensions, then the decoder reconstructs the data to resemble the input. The encoder and decoder compromise multiple Learnable Activation (LA) layers, each employing a set of learnable activation functions. Empirical studies on MNIST and Fashion-MNIST demonstrate better performance than Linear / Conv / LA encoders and Linear / LA decoders on the same latent dimension level. Reconstructed images by KARA are presented and compared to the original images.

**Strengths:**

1. The report on employing Learnable Activations to high-dimensional data processing is favoured.

**Weaknesses:**

1. Empirical studies on MNIST and Fashion-MNIST do not match the title 'Enhancing High-dimensional Data Processing' title and significantly lack sufficient empirical studies to support the superiority of KARA. Experiments covering CIFAR-100 and ImageNet and comparisons to SOTAs are expected.
2. The novelty and contributions to the field are questionable when the content is limited to the employment of Kolmogorov-Arnold theorem on auto-encoder architectures and discussion on shift-invariance and sparsification. Either theoretical analysis towards KARA's properties or sufficient experiments are essential.
3. No detailed experiment setup is provided in the paper (appendix).

**Questions:**

1. Can you discuss the computational perspective of KARA when comparing it to MLP, e.g. training/computational time, floating-time operations, time complexity, etc?
2. Can you discuss whether KARA can be applied to other tasks like NLPs?

---

### Official Review · Reviewer_owDs · 2024-11-04

**Soundness:** 1
**Presentation:** 2
**Contribution:** 1
**Rating:** 3
**Confidence:** 3

**Summary:**

This paper introduces the Kolmogorov-Arnold Representation Autoencoder (KARA), a novel autoencoder type. KARA makes use of learnable activation functions, leveraging the Kolmogorov-Arnold representation theorem. The architecture incorporates  a Learnable Activation (LA) layer and a Shift-Invariant Learnable Activation (SILA) layer, inspired to convolutions. These layers enable KARA to adjust to the data, enhancing model flexibility. Experimental results on datasets like MNIST and Fashion-MNIST show that KARA surpasses traditional models in data compression and image reconstruction tasks, and the authors suggest its usefulness for complex data modeling in various applications.

**Strengths:**

The paper is written in a plain, easy to read style. The organization of the material is rational, the mathematical section contains good explanations and and the line of reasoning is apparently easy to follow.
The integration of the Kolmogorov-Arnold representation theorem with neural networks, and more specifically in autoencoder architectures is interesting and underexplored.

**Weaknesses:**

While the quality of the work is commendable, the paper is limited by a lack of substantial experimentation, as it is confined to too simple datasets and architectures. The soundness of the results falls short of the level expected for a high-impact conference like ICLR, making it feel more like a promising proof of concept rather than a robust, field-advancing contribution.

This limitation is also reflected in the insufficiently supported claims. For instance, while since from the abstract it is stated that "KARA enhances the representational power and efficiency of neural networks, enabling superior performance in data compression tasks," the text does not characterize quantitatively these aspects—representational power, efficiency (under which point of view?), and compression—nor does it offer a clear comparison with a **fair** alternative approach, i.e., using the same (or a comparable) number of parameters.

**Questions:**

What are the results of a comparison between KARA and "standard" architecture with exactly the same number of learnable parameters?

**Details Of Ethics Concerns:**

No ethical concerns

---

### Comment · Area_Chair_WcXw · 2024-11-13
**authors - reviewers discussion open until November 26 at 11:59pm AoE**

Dear authors & reviewers,

The reviews for the paper should be now visible to both authors and reviewers. The discussion is now open until November 26 at 11:59pm AoE.

Your AC

---

### Note · Authors · 2025-01-21

I have read and agree with the venue's withdrawal policy on behalf of myself and my co-authors.